# An Open-Label Trial Study of Quality-of-Life Assessment in Irritable Bowel Syndrome and Their Treatment

**DOI:** 10.3390/medicina58060763

**Published:** 2022-06-05

**Authors:** Bogdana Ariana Alexandru, Lavinia Alina Rat, Andrada Florina Moldovan, Petru Mihancea, Lavinia Mariș

**Affiliations:** 1Faculty of Medicine and Pharmacy, Medicine Department, University of Oradea, 410068 Oradea, Romania; ariana.bogdana@gmail.com (B.A.A.); onita.andrada@yahoo.com (A.F.M.); 2Faculty of Medicine and Pharmacy, Doctoral School, University of Oradea, 1 December Square, 410068 Oradea, Romania; lavirat@yahoo.com (L.A.R.); petru.mihancea@yahoo.com (P.M.)

**Keywords:** irritable bowel syndrome, neurotransmitters, probiotics, quality of life

## Abstract

*Background and objectives:* Irritable bowel syndrome (IBS) does not only have a complex pathophysiological evolution with central and peripheral mechanisms. This study aimed to monitor the neuropsychiatric part of IBS and its management, following the quality of life of patients with IBS. *Materials and Methods:* Participants numbering 145 were investigated in this study for 6 months and were divided into four groups, namely the control group with a symptomatic period of less than 6 months (*n* = 34), the group with irritable bowel syndrome (IBS; *n* = 58), IBS and neuropsychiatric treatment (*n* = 32), and IBS with probiotic treatment (*n* = 22). Clinical and paraclinical analyses as well as quality of life were monitored by domestic and international psychological questionnaires. *Results:* It was observed that, in patients with pro-longed symptoms, neuropsychiatric impairment occurred more frequently, and both clinical and paraclinical analyses improved significantly (*p* < 0.05) more so in those with complex allopathic treatment and in those with probiotic treatment. There were no significant differences between the two research groups. *Conclusions:* It has been shown that the neuropsychological component of IBS plays an important role in its treatment, and modern probation therapy can achieve similar results to those of neuropsychiatry. This also requires further studies to ensure the best combination in the approach to IBS.

## 1. Introduction

Several central and peripheral mechanisms are involved in the pathophysiology of irritable bowel syndrome (IBS) [1]. The factors that affect the prevalence of IBS include food consumed, dysmotility, visceral hypersensitivity, post-infectious status, alterations in the intestinal microbiota, genetic factors, and psychosomatic factors [2,3]. Motility disorders are due to an increased frequency of bowel irritation, irregular contractions, prolonged transit time, and excessive motor response to food consumption. Serotonin plays an important role in regulating bowel motility, tenderness, and secretions [4,5].

Serotonin activates both intrinsic and extrinsic primary afferent neurons to initiate peristaltic reflexes and secretions and transmit signals to the central nervous system (CNS) [6]. The results of some studies have shown that in gastrointestinal functional abnormalities, including IBS, there are abnormalities in the serotonergic enteric signals that lead to hyperalgesia [6,7]. Changes in the serotonin signal cause enteric and extra-enteric symptoms. In the case of IBS where there predominantly appears to be diarrhea (IBS-D), there is a decrease in serotonin reuptake, while in the case of IBS where constipation (IBS-C) predominantly appears, there is an interruption of the serotonin signal [8]. Thus, the use of exogenous serotonin therapy agents may be considered in IBS therapy to provide optimal serotonin signals [9,10]. The effects of food are being followed by several studies. The findings of one study indicated that approximately 64–89% of patients with IBS reported complaints after eating certain foods. Persistent pain affects the quality of life, developing mental disorders [11,12].

The most common mental disorders that occur in IBS are represented by mood disorders (e.g., depression and dysthymia), anxiety disorder, and somatization disorder [13]. Among these disorders, depression is the most prevalent, which is also the most studied mental disorder in functional digestive disorders, with an incident rate of 31.4% in the case of IBS [14]. The somatization disorder accounts for a prevalence of 25% among patients with IBS. Highly varied symptoms include marked asthenia, headache, myalgia, and sexual dynamics disorders, which are followed by consequences such as a considerable number of medical consultations, the high costs of paraclinical examinations, and the frequent change of attending physicians [15].

More than half of the patients with IBS suffer from anxiety and/or depression, and the main peculiarity of patients with IBS-D is that psychiatric symptoms appear from the onset of the disease, compared to other subtypes of IBS [16]. The prevalence of anxiety disorder is reported at 15.5%, which is an important factor that influences a response to therapy and the severity of symptoms. Other common symptoms include insomnia, loss of appetite, and marked asthenia. The recognition of psychiatric symptoms is difficult and requires close collaboration among a family physician, gastroenterologist, internist, psychiatrist, and psychologist.

Panic attacks occur in approximately 12% of patients with IBS. The difficulty of correct diagnosis and effective treatment is due to the fact that somatic symptoms interact with psychological status, resulting in increased anxiety and depression. Data from the literature indicate that there is an association between people who have been sexually abused and IBS, accounting for about 40% of such cases.

Neuropsychological problems are common in people with IBS; for example, an increased likelihood of PTSD has been observed [17]. This supports a holistic consideration of the psychosocial aspects of IBS and further research into effective multi-modal therapeutics. The main objective of this study was to follow IBS from the earliest symptoms to prevent complicated forms. Psychological follow-up of patients and neuropsychiatric complications, as the most important disorders related to IBS, were also followed. This research aimed to look at the neuropsychiatric part of IBS as a factor that reduces the rate of improvement in symptoms, delays the process of healing, and affects the quality of life of patients with IBS.

Patients with IBS had significantly higher levels of anxiety and depression than healthy controls. Neuropsychiatric symptoms should be systematically checked and undergo treatment in patients with IBS, since psychological factors are important moderators of symptom severity, symptom persistence, and delayed healing.

Probiotic treatment has been widely studied in the last decade. The gut microbial flora is based on a wide variety of beneficial bacteria, fungi, and viruses. The human intestinal microbial flora has over 1000 species of commensal bacterial species. There are four major phyla’s that predominate in a healthy human gut: *Bacteroides, Firmicutes, Proteobacteria*, and *Actinobacteria*. These four bacterial species make up more than 90% of the microbiome’s bacterial population [18,19,20].

Short-chain fatty acids, including acetate, propionate, and butyrate, a bacterial metabolite, are synthesized in the bowels. Butyrate exhibits a wide range of pharmacological activities, including microbiome modulator, anti-inflammatory, anti-obesity, metabolic pathways regulator, anti-angiogenesis, and antioxidant [21,22].

*Firmicutes* and *Bacteroides* are the main butyrate that produces phyla. The most prominent groups of butyrogenic bacteria are *Faecalibacterium prausnitzii*, *Butyrivibriocrossotus*, and *Roseburia intestinalis*, which several studies have reported that the depletions of these bacteria, including atherosclerosis [23] and short-chain fatty acid producing bacteria (*Eubacterium rectale, Faecalibacterium prausnitzii, Akkermansia*, and *Bifidobacterium*), in symptoms of anxiety are common in chronic diseases such as IBS [24].

A high abundance of proteobacteria and low abundance of *Bifidobacteria* were observed in people with depression [25,26]. Low levels of neurotransmitters such as serotonin, dopamine, and norepinephrine can lead to depression. Various bacteria, such as *Streptococcus* spp., *Enterococcus* spp., *Escherichia* spp., *Lactobacillus plantarum*, *Klebsiella pneumonie*, and *Morganella morganii* have the ability to produce serotonin [27].

The human intestine is the largest molecular gateway in terms of digestion, decomposition of substances, and the formation of bioactivated molecules through the activity of the microbiome. It has been found that the presence of beneficial bacteria reduces responses to stress and anxiety and depressive behavior and promotes social behavior, decreases repetitive behavior, and improves cognitive function and communication in animals. Gastrointestinal comorbidities and food allergies are common in neurological disorders, suggesting a role for the gut microbiome [28].

A hypothesis developed in 2018 claims that depression, or the complications of this disorder, is a microglial disorder, because the onset of depression often follows intense inflammatory episodes in the brain and leads to a decline in microglial function. The gut microbiota has been found to communicate with the brain through several different mechanisms. This includes the production of neurotransmitters or the modulation of host neurotransmitter catabolism, innervation through the vagus nerve, or activation of the hypothalamic–pituitary axis [29].

## 2. Materials and Methods

The duration of the study has been mentioned as 6 months, following 145 patients with specific symptoms of IBS, using the Rome IV criteria, within 2019–2021. The patients were followed in the private medical office of Echo Laboratories, in Oradea, Romania. The specialized consultation was performed by the gastroenterologist and the psychological evaluation by the psychologist; the study is an open trial. The exclusion criterion was chronic inflammatory bowel diseases (e.g., Crohn’s disease and diverticulosis) or complicated IBS diagnosed before the start of the research study. Moreover, patients with gastroduodenal ulcers and gastrointestinal infections were excluded from the study. The treatment applied to all participants was performed according to the protocol based on personalized symptomatic therapy, including loperamidum, alverine with simeticonum, and a 6-strain probiotic complex with fructo-oligosaccharides containing Lactobacillus plantarum. Neuropsychiatric treatment is a cognitive-behavioral therapy and involved an inhibitor of serotonin reuptake, sertralinum, and, in the case of anxiety, lorazepamum. Probiotic therapy was based on the administration of probiotics based on *Bifidobacterium longum.*

The patients were divided into four groups, namely the control group (CG; *n* = 34), the group with an irritable intestine syndrome (IBS; *n* = 58), IBS and neuropsychiatric treatment group (IBS + NEURO; *n* = 32), and IBS and probiotic treatment (IBS + PROBIO; *n* = 22). The number of visits in the last month was taken as a specific indicator. The CG included those with the fewest visits. In the other 3 research groups, the open label trial study allowed the patient to choose whether to accept neuropsychiatric treatment, or probiotherapy, or only symptomatic treatment specific to IBS, having access to information related to treatment. All those who were included in the 3 research groups were present at least twice a month for control.

Each patient was present at the doctor’s office at least once a month, and evaluations were performed at the beginning and end of the research period.

### 2.1. Clinical Diagnosis

Diagnosis according to the current protocol was reached by eliminating other gastrointestinal diseases and monitoring gastrointestinal pain, stool pain, the period in which symptoms are present, stool consistency (i.e., diarrhea, mixed stools, or constipation), and changes in the appearance of stool.

### 2.2. Paraclinical Diagnosis

Paraclinical analyses were performed in the analysis laboratory by the enzymatic, colorimetric, and spectrophotometric methods and enzyme-linked immunosorbent assay. We monitored 4 metabolic parameters, namely cholesterol, triglycerides, uric acid, and C-reactive protein (CRP), at the beginning and end of the research period.

### 2.3. Questionnaires

The Montgomery-Asberg Depression Rating Scale (MADRS) is a depression-measuring questionnaire. This assessment is of particular importance due to the permanent presence of pain reported by patients with IBS.

The Montgomery–Asberg Depression Scale follows reported sadness, apparent sadness, inner tension, decreased sleep, decreased appetite, difficulty concentrating, exhaustion, inability to feel or rejoice, pessimistic thoughts, and suicidal thoughts. In this questionnaire, the evaluation scale is from 0 to 6, where “0” represents a normal state, without sadness or pessimistic thoughts and “6” represents a very sad state.

Hamilton Anxiety Assessment Scale (HAS) is designed to help psychiatrists and clinical psychologists quantify the symptoms of patients already diagnosed with neurasthenic anxiety. However, this scale was not developed to assess such conditions as depression, obsessive-compulsive disorder, organic dementia, or schizophrenia. This instrument is often used for a range of mental illnesses, where anxiety is the predominant symptom.

The Hamilton Anxiety Rating Scale looks for anxiety, anticipation of upset, frightening premonitions, irritability, trembling, fear of the dark, interrupted sleep, nervous tics, twitching, muscle twitching, gnashing of teeth, insecure voice, difficulty swallowing, meteor restlessness, agitation, restlessness or placidity, trembling hands, frowning, and tense face. In this questionnaire, also, the evaluation scale is from 0 to 4, where “0” is the absence of symptoms and “4” represents severe symptoms.

The Hamilton Depression Scale Questionnaire (HDS) is employed to assess depression, the degrees of depression, and the impact of depression degrees on the quality of life.

The Hamilton Depression Scale tracks depression, feelings of guilt, the idea of “suicide,” difficulty in work and activities, retardation, mental anxiety (psychiatric anxiety), somatic anxiety, genital symptoms, hypochondria, and self-analysis. The interpretation of this questionnaire is made by adding the points where <7 is without depression, 7–17 represents mild depression, 18–24 represents moderate depression, >25 represents severe depression.

### 2.4. Statistical Analysis

The obtained data were analyzed in the statistical program SPSS 20 (New York, NY, USA) through statistics (ANOVA), Chi-square test, and inferential statistics (Student t-test), and the 4 research groups were compared with the Bonferroni post hoc tests. The correlations between the parameters were verified using Bravais–Pearson tests and with paired sample correlation.

### 2.5. Statement

This study was carried out in agreement with the research ethics committee within the Faculty of Medicine and Pharmacy, University of Oradea, Oradea, no. 12/01.04.2019, as well as all approvals for quality-of-life questionnaires, translated into the research language, Romanian.

## 3. Results

### 3.1. Demographic Description

A total of 146 individuals (76 men and 70 women) were included in the research study and divided into 3 groups of GC (*n* = 34), IBS (*n* = 58), IBS + NEURO (*n* = 32), and IBS + PROBIO (*n* = 22), representing 23.3%, 39.7%, 21.9%, and 15% of all participants, respectively. Regarding the place of residence, 117 and 29 subjects, respectively, were from urban and rural areas (respectively, Chi-square test χ^2^ = 53,041 and *p* = 0.001), significantly more participants from urban areas. In CG (*n* = 34), 18 cases were male and 16 individuals were female. The IBS group (*n* = 58) consisted of 24 men and 34 women. The IBS + NEURO group (*n* = 32) consisted of 22 men and 10 women and 12 men with 10 women in IBS + PROBIO. After statistical processing with the Chi-square test, we obtained χ^2^ = 0.236 (*p* = 0.619) for the cohort, i.e., statistically insignificant differences and without significant difference in groups: χ^2^ = 5.198, *p* = 0.072.

Table 1 presents the demographic characteristics of the participants in the research study.

In the entire cohort, the youngest and oldest patients were 20 and 75 years old, respectively, and the mean age was obtained at 39.99 ± 12.26 years.

The distribution of patients in the four groups was performed in a similar manner. Accordingly, the mean age scores in the CG, IBS, IBS + NEURO, and IBS + PROBIO groups were estimated at 38.71 ± 14.38, 43.62 ± 13.13, 34.78 ± 8.66, and 40.00 ± 7.36 years, respectively.

#### 3.1.1. Clinical Data

According to Rome IV criteria, patients with IBS are defined as those who have such symptoms as recurrent abdominal pain manifested at least 1 day in a week in the last 3 months and who are associated with two of the following characteristics:The pain is related to defecation;The onset of symptoms is associated with changes in the frequency of stool;The onset of symptoms is associated with a change in the shape or appearance of the stool.

Among the specific aspects of IBS, the researcher followed the pain as a decisive symptom in the patient’s management. Whether or not the pain is related to defecation, as long as the symptoms are present such as abdominal pain, the number, shape, and appearance of the altered stool are part of the diagnosis of IBS.

According to Table 2, pain was present in all patients regardless of the group in which they were included. The highest percentage of pain related to the defecation process was reported for 94.64% of the patients in the IBS group (χ^2^ = 82,877, *p* = 0.001), followed by 78.12% and 86.09% of the subjects in the IBS + NEURO and IBS + PROBIO groups, respectively, and 90.98% in the CG group.

The control group showed all symptoms of IBS less than 3 months, while in the other three research groups, the symptoms were present for over 6 months in 55.17% of patients in the IBS group and 57.40% of the individuals in the IBS + NEURO group. The presence of symptoms for 3–6 months was observed in 39.7% and 22.0% of the cases in the IBS and IBS + NEURO groups and 15.1% in the IBS + PROBIO group, respectively. In the cohort, the most frequent duration of symptoms was reported for more than 6 months in 43.2% of the participants (χ^2^ = 8644, *p* = 0.013).

Irritable bowel syndrome is characterized by mixed stools (alternatively constipation and diarrhea) or by multiple stools per day (over three stools), which was also observed in the present study. In this regard, 2.94% in CG, 6.2% in IBS, and 2.7% in the IBS + NEURO and in the IBS + PROBIO of the patients suffered from constipation. It was found that 10.3% in IBS, 6.2% and 4.1% of the cases in the IBS + NEURO and IBS + PROBIO groups had mixed stools. Moreover, 7.5% in CG, 23.3% in IBS, 13.0% in IBS + NEURO, and 8.2% in the IBS + PROBIO presented with more than three stools per day. Regarding the cohort, 52.1% of the subjects presented with over three stools per day (χ^2^ = 34.904, *p* = 0.001).

The appearance of the stool changed in 81.50% of the patients, rendering a coefficient of χ^2^ = 57,973 (*p* = 0.001). This change was confirmed in 76.40% of the cases in the control group, 84.48% of the subjects in the IBS group, 81.25% of the patients in the IBS + NEURO group, and 81.81% of the patients in the IBS + PROBIO group.

#### 3.1.2. Paraclinical Data

Cholesterol, triglycerides, uric acid, and CRP were followed paraclinically. To determine the effect of the type of intervention on cholesterol, we used the Chi-square statistical test and obtained χ^2^ = 11,193 (*p* = 0.003); regarding this, in the post-test (at the end of the research period), cholesterol differed significantly, as is summarized in Table 3.

Considering triglycerides, we employed the Chi-square statistical test for the two independent groups and obtained χ^2^ = 13,372 (*p* = 0.001); therefore, in the post-test (after 6 months), the triglycerides showed significant differences, as observed in Table 3. There was a statistically significant decrease in the IBS group, compared to CG.

Following uric acid, we applied the Chi-Square and Wilcoxon Signed Ranks test for the two independent groups and obtained χ^2^ = 3453 (*p* = 0.178); accordingly, in the post-test (after 6 months), uric acid did differ significantly in the two groups, as shown in Table 3.

Regarding the verification of the effect of the intervention type on CRP, we used the Chi-square statistical test and obtained χ^2^ = 10,044 (*p* = 0.001); regarding this, in the post-test (after 6 months), CRP was significantly different, as observed in Table 3.

The differences were obtained and processed graphically by error bar figures. In Figure 1, the biggest differences were observed in the IBS group, presented by cholesterol, but triglycerides, uric acid, and CRP, and the biggest differences were observed in the IBS + NEURO group. The differences were obtained and graphically processed by the error bar numbers. In Figure 1, the largest differences were observed in the IBS group, shown by cholesterol, but for triglycerides, uric acid, and CRP, the largest differences were observed in the IBS + NEURO group.

Significant differences between IBS + NEURO and IBS + PROBIO can be observed only in the case of uric acid; otherwise, no significant differences are observed between the two research groups.

The final comparative results with the initial data in this subchapter are presented in Table 4, according to which a significant change can be observed following the coefficient of Paired Samples Correlations (r) and Wilcoxon Signed Ranks.

#### 3.1.3. Assessing the Quality of Life

The quality of life was evaluated using three questionnaires, namely MADRS, HAS, and HDS.

The graphical presentation of MADRS in each group at the beginning of the study can be seen in Figure 2. It depicts the lowest values in the CG, meaning that the symptomatology period greatly influenced the quality of life, which was measured using MADRS. In the IBS + NEURO group, following the neuropsychiatric treatment, more than 5% of the patients did not present any symptoms of depression on this scale. The best results were observed in the IBS + PROBIO group.

At the end of the study (at 6 months), compared to the initial assessment, the following results were observed:In CG, the number of asymptomatic patients increased by 0.7%, mild depression decreased by 3.5%, moderate depression did not change, and severe depression decreased by 0.7% of cases. These results are correlated with the increased frequency of PTSD in those with IBS. No significant differences were obtained in this group comparing the final results with the initial values.In the IBS group, no change was observed in the number of asymptomatic patients, mild depression decreased by 2.0%, moderate depression decreased by 1.4%, and severe depression increased by 3.5% of cases.In the IBS + NEURO group, the number of asymptomatic patients increased by 10.3%, mild depression increased by 6.9%, moderate depression did not change, and severe depression decreased by 8.2% of cases. At the end of the study, a significant decrease in incidence in this group with severe depression measured on the MADRS scale was observed.

Figure 3 shows that none of the patients in GC are without anxiety. This can be explained by the fact that each person reacts to persistent pain with a medium or high intensity. In cases of prolonged pain, after the pain subsides, the fear of pain appears.

Following statistical processing, regarding the differences between the final and the initial values, we applied the One-Sample test on the four independent groups and obtained t = 4.437 (*p* = 0.001), which was statistically significant. The results of the Bonferroni post hoc test on the two independent variables showed significant differences between the CG and IBS groups (*p* = 0.004), the CG and IBS + NEURO groups (*p* = 0.001), and the IBS and IBS + NEURO groups (*p* = 0.001). *p* = 0.050), and between CG and IBS + PROBIO and IBS + NEURO and IBS + PROBIO, the differences are insignificant (*p* > 0.05).

At the end of the study (at 6 months), compared to the initial assessment, the following findings were reported:In CG, the number of asymptomatic patients did not change, mild anxiety increased by 0.6%, moderate anxiety did not change, severe anxiety increased by 4.8%, and very severe/severe anxiety decreased by 5.5% of cases. Due to the symptomatic period of less than 3 months, at the end of the research period (6 months), the symptoms have improved and the quality of life improved significantly (*p* = 0.001).In the IBS group, no change was observed in the number of asymptomatic patients, mild anxiety decreased by 8.0%, moderate anxiety decreased by 6.0%, severe anxiety increased by 14%, and very severe/severe anxiety did not change in cases.In the IBS + NEURO group, the number of asymptomatic patients increased by 12.0%, mild anxiety increased by 12.0%, moderate anxiety increased by 11.0%, severe anxiety decreased by 6.0%, and very severe/severe anxiety decreased by 7%. At the end, no person in this group was reported with very severe anxiety measured with the HAS scale.

Following statistical processing, regarding the differences between the final and the initial values, we applied the One-Sample test on the four independent groups and obtained t = 4.130 (*p* = 0.001), which was statistically significant. Based on the results of the Bonferroni post hoc test on the two independent groups, the comparison of the CG with the IBS group showed significant differences (*p* = 0.045), while there were insignificant differences between the CG and IBS + NEURO groups (*p* = 0.176) and between the IBS and IBS + NEURO groups (*p* = 1000), as presented in Figure 4.

Following statistical processing, regarding the differences between the final and the initial values, we applied the One-Sample test on the four independent groups and obtained t = 2.582 (*p* = 0.011), which was statistically significant. The results of the Bonferroni post hoc test on the two independent groups revealed significant differences in comparing the CG with the IBS group (*p* = 0.001), the CG with the IBS + NEURO group (*p* = 0.006), the IBS group with IBS + NEURO group (*p* = 0.001), the CG with the IBS + PROBIO group (*p* = 0.040), and IBS with all groups (*p* < 0.05), as depicted in Figure 4.

At the end of the study (at 6 months), compared to the initial assessment, the following results were obtained:In CG, the number of asymptomatic patients increased by 2.1%, mild depression decreased by 2.1%, moderate depression increased by 0.7%, and severe depression decreased by 0.7% of cases, but without reaching the lower limit of statistical significance.In the IBS group, there was a 0.7% decrease in the number of asymptomatic patients, a 12.6% decrease in mild depression decreased, no change in moderate depression, and a 10.2% increase in severe depression in cases.In the IBS + NEURO group, the number of asymptomatic patients increased by 13.1%, mild depression increased by 2.8%, moderate depression decreased by 6.1%, and severe depression decreased by 9.6% of cases, measured with the HDS scale, compared with IBSIn the IBS + PROBIO group, the number of asymptomatic patients decreased by 2.9%, mild depression increased by 11.1%, moderate depression decreased by 5.2%, and severe depression decreased by 3.2% of cases, measured with the HDS scale, compared with IBS + NEURO.

#### 3.1.4. Correlations

As MADRS depression increased, it affected the quality of life, which consequently led to a rise in HAS; it shows a strong relationship. It was found that MADRS was also positively correlated with the HDS scale. An increase in depression followed by the MADRS scale resulted in a rise in depression measured with the HDS scale.

#### 3.1.5. Comparison of IBS-NEURO and IBS-PROBIO

Following the two research groups, no statistically significant differences were observed between the two groups, and better results were observed for neuropsychiatric treatment, but with insignificant differences, which are presented in Table 5.

#### 3.1.6. Linear Regression

Pearson’s correlation shows that there is a strong link between the two parameters, i.e., in the control group and the IBS group, and it appears that the quality of life decreased. In the IBS + NEURO and IBS + PROBIO groups, a reduction in depression and anxiety and an increase in quality of life were observed. The estimation of depression in the first research groups (CG and IBS) is represented by R2 shows that 94.8% of the group is involved with MADRS, 91.8% with HAS and 91.2% with HDS, respectively (Table 6 and Figure 5). The ANOVA test shows whether the model is statistically relevant.

The estimate for the parameter is significant, which shows the value of the standardized and the non-standardized coefficient as well as the value of statistical significance *p* < 0.05.

## 4. Discussion

Irritable bowel syndrome is a functional gastrointestinal disorder with symptoms, including abdominal pain, associated with a change in stool shape or frequency [30]. The pathophysiology of IBS is not yet completely understood; however, it is well established that there is disordered communication between the intestines and the brain, leading to motility disorders, visceral hypersensitivity, and altered CNS processing [30]. In our study, the syndrome was followed from several points of view, focusing on the neuropsychiatric connection, which has often been neglected. 

The effective management of IBS, a common functional gastrointestinal disorder, can be challenging for physicians due to the lack of simple diagnostic tests and a wide variety of available treatment approaches [31]. For this reason, the clinical symptoms of gastrointestinal disorders (i.e., pain, defecation-related pain, constipated or mixed diarrhea, and changes in the appearance of the stool) and the duration of the presence of these symptoms were followed.

Changing eating habits permanently with the application of a healthy diet is most difficult in patients with IBS. In studies on IBS, the basis is the implementation of a healthy lifestyle [32,33]. In several studies, the researcher observed a reduction in symptoms as a result of following a healthy diet [34,35]. Many studies have been published about the beneficial effects of the Mediterranean diet as a basis for a healthy diet [36], with an important role in the management of cardiovascular diseases, obesity, or hypertension. Other studies have shown the effectiveness of the low fodmap diet [32], but, most importantly, vitamin D supplementation in the management of IBS is effective [37]. These diets have been followed in various meta-analyses, and it has been found that the diet helps reduce symptoms, and it positively influences the quality of life [38,39,40,41,42]; however, the studies have many unanswered questions, so further studies are needed. The results of a healthy diet have been reflected in paraclinical analyses, where a significant improvement was observed in the two research groups, while in the control group, no significant values were modified.

Gastrointestinal pain is characteristic of this syndrome; in this respect, the management of patients with IBS is a challenge since diagnosis and treatment may require several approaches with unsatisfactory outcomes. In any case, the diagnosis of IBS is based on the positive identification of symptoms in accordance with this condition and by the exclusion of a minor organic disease [43].

In 2019, IBS was defined as a complex of chronic, recurrent symptoms, which included abdominal pain or discomfort, changes in bowel habits, and bloating for at least 6 months, which was somewhat broader than the previous definition [44].

The links of IBS to neuro-mental problems, such as depression and anxiety, have been discussed in specialized studies [45,46]; nevertheless, these have been discussed for the rebalance of deficient microbiota [47]. Dysbiosis has also been associated with other chronic diseases (e.g., type 2 diabetes) [18,48] and periodontal disease [49]; these diseases are related to metabolic syndrome [50,51,52]. From this point of view, we followed five psychological parameters in order to correlate with the evolution of IBS. These parameters were tracked to examine if probiotics improved mood, stress, and anxiety in a selected sample for low mood. We also tested whether the presence or severity of symptoms of IBS and the levels of proinflammatory cytokines, brain-derived neurotrophic factor, and other blood markers would predict or influence response to treatment [53].

The quality of life in patients with IBS is affected due to persistent gastrointestinal pain, which is followed after special diets [54]. In 2016, Bian followed IBS by the severity and quality of life (QoL) score. A moderate linear correlation was observed [55]. Probiotics are widely used as alternative therapies, observing the effect on health by preventing neuropsychiatric complications, as is shown by the results of a study conducted in 2018 [56]. In the current study, the follow-up and evolution of the quality of life was part of the complex follow-up of IBS. 

There has been a link between high levels of norepinephrine and IBS [26] and elevated levels of norepinephrine in the absence of *Firmicutes* [26]. In our study, probiotics administered on the basis of butyrenic firmicute contributed to the improvement of the quality of life by reducing depression and anxiety.

One of the limitations of this study was primarily related to the existence of multiple intestinal diseases, which within 6 months might have coincided with the followed symptoms. Another limitation was the infection with the SARS-CoV-2 virus, which left patients with neurological issues and, consequently, might have affected the results and would require the follow-up of patients in this regard. The minimum period of the presence of symptoms was 6 months, which did not allow tracking the cyclicity of the disease in the study period; therefore, it requires additional studies.

Patients with long-term symptoms diagnosed with IBS and those who fail to change their diet profoundly are more affected. Although the symptomatic treatment has been shown to be effective, it is insufficient; on the other hand, better results may be obtained with neuropsychiatric treatment.

## 5. Conclusions

In the CG group, no improvements in paraclinical parameters and quality of life were observed, and there were even worsening effects due to associated PTSD.

Stool pain, as a clinical symptom, was 2.1% more present in the IBS group than in the IBS + NEURO or IBS + PROBIO group.

In the case of both the administration of neuropsychiatric allopathic treatment and the administration of probiotics, the clinical and paraclinical results and the quality of life significantly improved. Neuropsychiatric allopathic treatment has been shown to be the most effective in reducing symptoms and contributing to increased quality of life but with insignificant differences from the probiotic treatment group.

This finding highlights the possibility of replacing neuropsychiatric treatment with probiotic therapy associated with IBS-specific allopathic treatment, thus reducing the neurological adverse effects of treatments (drowsiness, dizziness, and addiction).

## Figures and Tables

**Figure 1 medicina-58-00763-f001:**
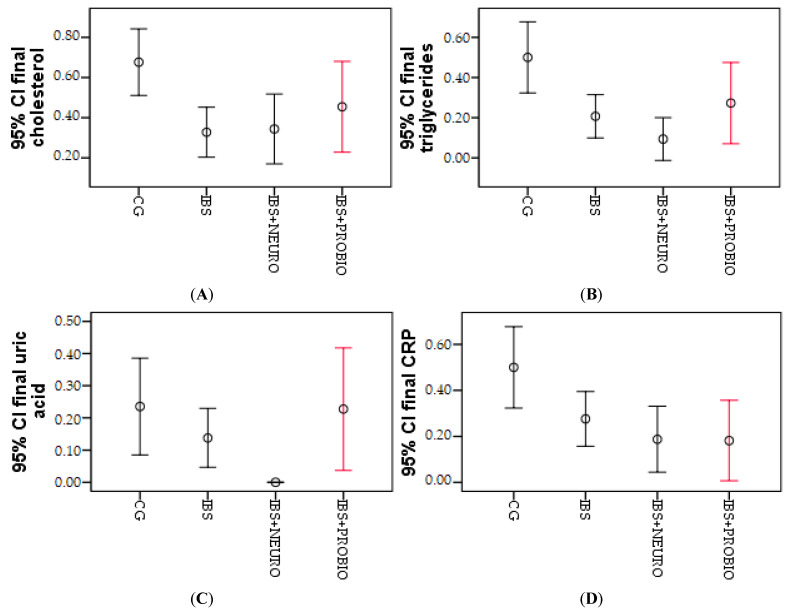
Graphical representation of the mean cholesterol (**A**), triglycerides (**B**), uric acid (**C**), and CRP (**D**) at the end of the research period in the 3 groups.

**Figure 2 medicina-58-00763-f002:**
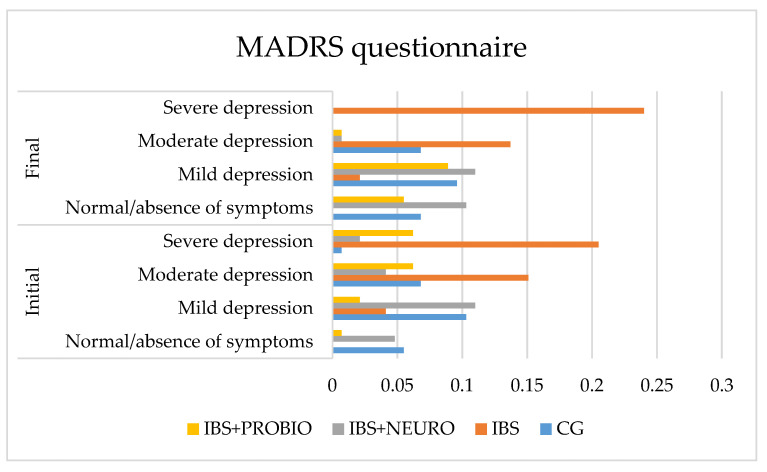
Graphical presentation of the MADRS in each group at the beginning and end of the study period.

**Figure 3 medicina-58-00763-f003:**
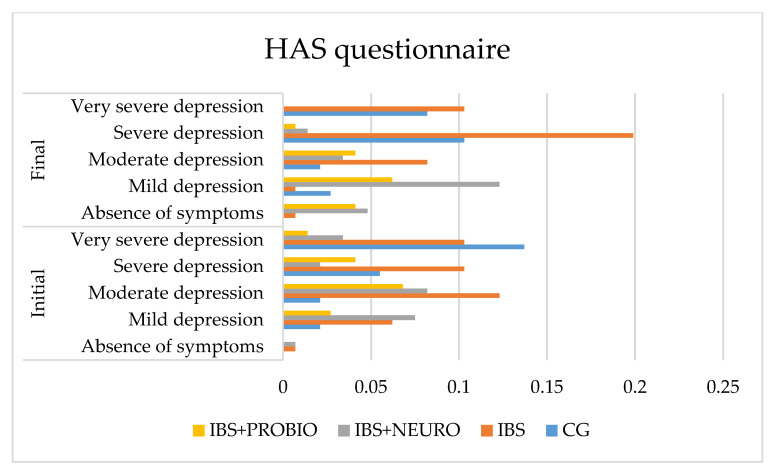
Graphical presentation of the HAS scale in each group at the beginning and end of the study period.

**Figure 4 medicina-58-00763-f004:**
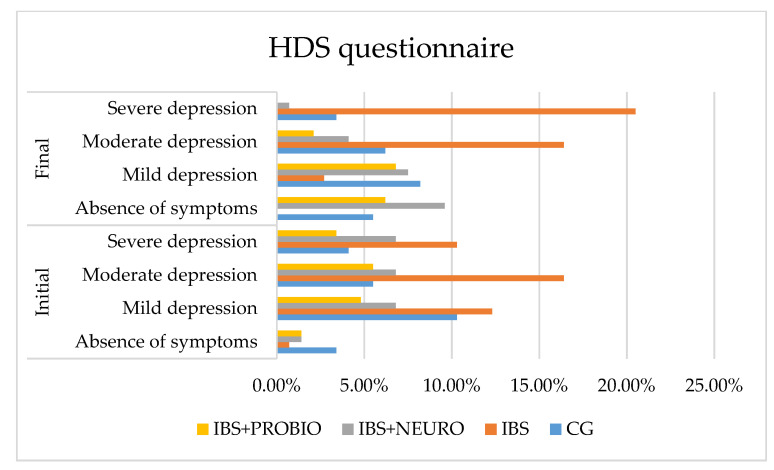
Graphical presentation of the HDS scale in each group at the beginning and end of the study period.

**Figure 5 medicina-58-00763-f005:**
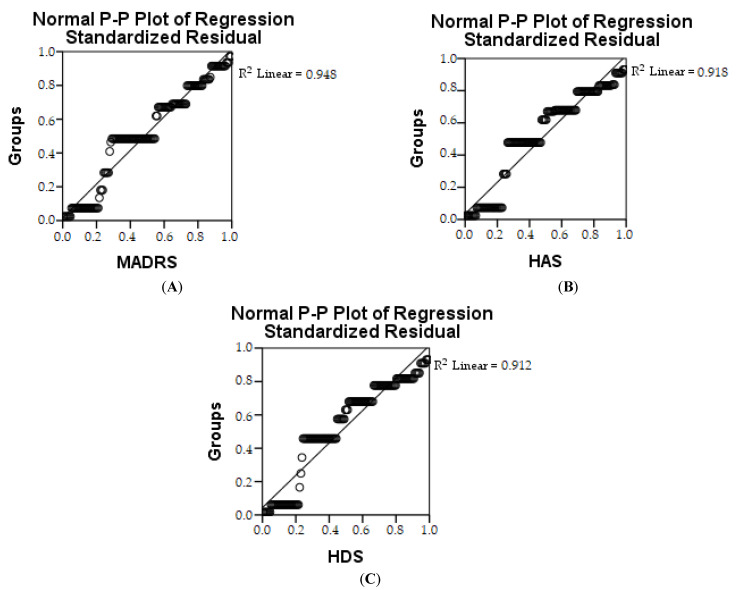
Linear regression of MADRS (**A**), HAS (**B**), and HDS (**C**) depending on groups.

**Table 1 medicina-58-00763-t001:** Demographic characteristics of the study and cohort participants.

Demographic Description		Groups								Total	
		CG		IBS		IBS + NEURO		IBS + PROBIO			
		Count	%	Count	%	Count	%	Count	%	Count	%
Gender	Male	18	12.3	24	16.4	22	15.1%	12	8.2%	76	52.1
	Female	16	11.0	34	23.3	10	6.8%	10	6.8%	70	47.9
Area	Urban	27	18.5	47	32.2	23	15.8%	20	13.7%	117	80.1
	Rural	7	4.8	11	7.5	9	6.2%	2	1.4%	29	19.9

CG = Control group. IBS = Group with an irritable intestine syndrome. IBS + NEURO = Group with an irritable intestine syndrome and neuropsychiatric treatment. IBS + PROBIO = Group with an irritable intestine syndrome and probiotic treatment.

**Table 2 medicina-58-00763-t002:** Description of initial clinical characteristics of the patients in the 4 research groups.

Parameters		Groups								Total	
		CG		IBS		IBS + NEURO		IBS + PROBIO			
		Count	%	Count	%	Count	%	Count	%	Count	%
Pain	Absent	0	0.0	0	0.0	0	0.0%	0	0.0%	0	0.0
	Present	34	23.3	58	39.7	32	21.9%	22	15.1%	146	100.0
Pain related to the defecation	Absent	3	2.1	5	3.4	7	4.8%	3	2.1%	18	12.3
	Present	31	21.2	53	36.3	25	17.1%	19	13.0%	128	87.7
Presence duration of symptoms	Less than 3 months	34	23.3	0	0.0	0	0.0%	0	0.0%	34	23.3
	Between 3 and 6 months	0	0.0	26	17.8	16	11.0%	7	4.8%	49	33.6
	More than 6 months	0	0.0	32	21.9	16	11.0%	15	10.3%	63	43.2
Stool consistency	Constipation	1	0.7	9	6.2	4	2.7%	4	2.7%	18	12.3
	Mixed	22	15.1	15	10.3	9	6.2%	6	4.1%	52	35.6
	Over 3 stools a day	11	7.5	34	23.3	19	13.0%	12	8.2%	76	52.1
Changes in the appearance of the stool	No	8	5.5	9	6.2	6	4.1%	4	2.7%	27	18.5
	Yes	26	17.8	49	33.6	26	17.8%	18	12.3%	119	81.5

CG = Control group. IBS = Group with an irritable intestine syndrome. IBS + NEURO = Group with an irritable intestine syndrome and neuropsychiatric treatment. IBS + PROBIO = Group with an irritable intestine syndrome and probiotic treatment.

**Table 3 medicina-58-00763-t003:** Descriptive statistics for the comparison of mean paraclinical indicators for the 4 research groups.

Variable	Groups								Total	
	CG		IBS		IBS + NEURO		IBS + PROBIO			
	Mean	SD	Mean	SD	Mean	SD	Mean	SD	Mean	SD
Cholesterol initial	0.68	0.47	0.72	0.45	0.44	0.50	0.68	0.48	0.64	0.48
Cholesterol final	0.68	0.47	0.33	0.47	0.34	0.48	0.45	0.51	0.43	0.50
Triglycerides initial	0.41	0.50	0.64	0.48	0.41	0.50	0.59	0.50	0.53	0.50
Triglycerides final	0.50	0.51	0.21	0.41	0.09	0.30	0.27	0.46	0.26	0.44
Uric acid initial	0.24	0.43	0.50	0.50	0.13	0.34	0.59	0.50	0.37	0.48
Uric acid final	0.24	0.43	0.14	0.35	0.00	0.00	0.23	0.43	0.14	0.35
CRP initial	0.50	0.51	0.50	0.50	0.25	0.44	0.55	0.51	0.45	0.50
CRP final	0.50	0.51	0.28	0.45	0.19	0.40	0.18	0.39	0.29	0.46

CG = Control group. IBS = Group with an irritable intestine syndrome. IBS + NEURO = Group with an irritable intestine syndrome and neuropsychiatric treatment. IBS + PROBIO = Group with an irritable intestine syndrome and probiotic treatment. CRP = C-reactive protein. SD = Standard deviation.

**Table 4 medicina-58-00763-t004:** Paired correlation of initial and final cholesterol, triglycerides, uric acid, and CRP.

Variable		Mean	SD	Lower	Upper	T	Sig.	r/Z	Sig.
1	Cholesterol initial—cholesterol final	0.21233	0.55347	0.12180	0.30286	4.635	0.001	0.359	0.001
2	Triglycerides initial—triglycerides final	0.26712	0.50229	0.18496	0.34928	6.426		0.436	
3	Uric acid initial—uric acid final	0.22603	0.41970	0.15738	0.29468	6.507		5.745	
4	CRP initial—CRP final	0.15753	0.36556	0.09774	0.21733	5.207		0.711	

SD = Standard deviation. T = Student’s *t*-test coefficient. Sig = Statistical signification. r = Paired samples correlations coefficient. CRP = C-reactive protein.

**Table 5 medicina-58-00763-t005:** Comparison of final results in the IBS + NEURO and IBS + PROBIO groups.

Variables		*n*	Mean	Std. Deviation	*p*
Cholesterol	IBS + NEURO	32	0.3438	0.48256	1.000
	IBS + PROBIO	22	0.4545	0.50965	
Triglycerides	IBS + NEURO	32	0.0938	0.29614	0.759
	IBS + PROBIO	22	0.2727	0.45584	
Uric acid	IBS + NEURO	32	0.0000	0.00000	0.112
	IBS + PROBIO	22	0.2273	0.42893	
CRP	IBS + NEURO	32	0.1875	0.39656	1.000
	IBS + PROBIO	22	0.1818	0.39477	
MADRS	IBS + NEURO	32	0.5625	0.56440	1.000
	IBS + PROBIO	22	0.6818	0.56790	
HAS	IBS + NEURO	32	1.0625	0.80071	1.000
	IBS + PROBIO	22	1.0909	0.86790	
HDS	IBS + NEURO	32	0.8125	0.85901	1.000
	IBS + PROBIO	22	0.7273	0.70250	

**Table 6 medicina-58-00763-t006:** Regression table of quality-of-life parameters according to research groups.

Parametri	Correlate	R^2^ (%)	r	β1	β2	t	F
MADRS	Groups	94.80	0.432 **	−0.379 **	−0.432 **	5.749 **	33.050 **
HAS		91.80	0.419 **	−0.372 **	−0.418 **	5.519 **	30.454 **
HDS		91.20	0.471 **	−0.398 **	−0.471 **	6.406 **	41.043 **

R^2^ = R square. β1 = unstandardized coefficient. β2 = standardized coefficient. t = coefficient of correlation. F = coefficient ANOVA, ** Correlation is significant at the 0.01 level (2-tailed).

## Data Availability

Data supporting the reported results can be found in the archives ECHO LABORATORIES for the years 2019–2021.

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
