# Peer review of "An Open-Label Trial Study of Quality-of-Life Assessment in Irritable Bowel Syndrome and Their Treatment"

_medicina, 2022, doi:10.3390/medicina58060763_

Round 1

Reviewer 1 Report

Although the topic is clinically relevant, the quality of the presentation is low and the manuscript requires extensive edits for language, clarity, and style.

Specific comments:

  1. Suggest shortening the abstract to 200 words as per the journal's guidelines.
  2. The study title is slightly misleading, suggest rewording this to an open trial.
  3. "The factors that affect the prevalence of IBS include food consumed, dysmotility, visceral hypersensitivity, post-infectious status, alterations in the intestinal microbiota, genetic factors, and psychosomatic factors [2]" - in addition to ref [2], suggest including a newer review highlighting the role of inflammation in the etiopathogenesis of IBS (citation: pubmed.ncbi.nlm.nih.gov/30288077).
  4. The introduction section is much too long with a lot of irrelevant details that should be trimmed. "The largest amount of serotonin is produced by enterochromaffin cells and acts as a signaling agent in the intestine, influencing enterocytes, smooth muscles, and the enteric nervous system [4]. Serotonin activates both the intrinsic and extrinsic primary afferent neurons to initiate peristaltic reflexes and secretions and transmit signals to the central nervous system (CNS) [5]" - I am not sure how these extended descriptions are relevant to the topic at hand.
  5. "The symptoms of IBS are correlated with a high rate of neuropsychological problems" - it would be relevant to say that there is also a significant association between PTSD and IBS (citation: pubmed.ncbi.nlm.nih.gov/30144372). This supports a holistic consideration of the psychosocial aspects of IBS and further research into effective multi-modal therapeutics.
  6. Scientific names such as "Bacteroides, Firmicutes, Proteobacteria and Actinobacteria" should be italicized as per convention.
  7. How was the sample size determined? There was no evidence of power calculations.
  8. "Diagnosis according to the current protocol" - did you use the Rome IV criteria? Please specify.
  9. "neuropsychic treatment" is misspelled. Do you mean "neuropsychiatric treatment"?
  10. What exactly does "neuropsychic treatment" entail? Is this some modified form of cognitive-behavioral therapy?
  11. There are simply too many figures and many of the figures can be omitted and placed under supplementary materials instead.
  12. "Changing eating habits permanently with the application of a healthy diet is most difficult in patients with IBS. In studies on IBS, the basis is the implementation of a healthy lifestyle [29,30]" - what constitutes a healthy diet? This is too vague. There isn't really a strong evidence base for elimination diets or the FODMAP diet as well. What about the role of vitamin supplementation? This should be discussed as well (citation: pubmed.ncbi.nlm.nih.gov/35396764).
  13. The way QoL is measured in previous IBS studies has lacked consistency and does not lend itself well to analysis. This is an important point to discuss.
  14. No discussion of study limitations.
  15. Please write the conclusion paragraph in continuous and coherent prose rather than point form.

Author Response

Response to Reviewer 1

Firstly, I, the author of the present manuscript wish to thank you for thoughtful commentary you have provided to improve the quality of the paper. I am very grateful for the time and effort you have devoted to this task. We have extensively revised my manuscript according to the recommendations. All changes in the text and the new figures that we have redesigned are highlighted. Please, see the point-by-point answers to your comments below. All correction was highlighted in the manuscript.

Reviewer 2 Report

the authors need to improve a description of the method and a statistical analysis.

the graphs must be re-presented to allow a comparison between before and after values, side by side.

The study has no description of the type and steps performed. I suggest to authors follow a guideline like STROBE or CONSORT to write the essential parts of the manuscript in a logical and reliable way.

The authors need to revise the statistical analysis and present then in a mature way. Column graphs without statistical comparison are not a good way. Besides that, ANOVA is not a descriptive analysis as the authors said in the method section.

The question of this research demands an association analysis, not only comparing means. Do the authors have social and clinical variables why not performed a regression analysis?

Author Response

Response to Reviewer 2

Firstly, I, the author of the present manuscript wish to thank you for thoughtful commentary you have provided to improve the quality of the paper. I am very grateful for the time and effort you have devoted to this task. We have extensively revised my manuscript according to the recommendations. All changes in the text and the new figures that we have redesigned are highlighted. Please, see the point-by-point answers to your comments below. All correction was highlighted in the manuscript.

Round 2

Reviewer 1 Report

1. Extensive edits for language are still required.

2. "... translated into the research language" - which language? Please specify.

3. Are the error bars in Figures 2 to 4 essential? They are all over the place.

Author Response

We are very grateful for the effort and time you have devoted to this task. We, the authors of the present manuscript wish to thank you for thoughtful commentary you have provided to improve the quality of the paper. We have extensively revised our manuscript according to the recommendations. All changes in the text and the new figures that we have redesigned are highlighted. Please, see the point-by-point answers to your comments below.

Reviewer 2 Report

The authors improved the quality of the manuscript and I have no more suggestions. 

Author Response

We are very grateful for the effort and time you have devoted to this task. We, the authors of the present manuscript wish to thank you for thoughtful commentary you have provided to improve the quality of the paper. 
